Iodine nutrition and toxicity in Atlantic cod (Gadus morhua) larvae

Penglase S 1 2
Harboe T 3
Sæle Ø 1
Helland S 4
Nordgreen A 1 5
Hamre K kha@nifes.no 1
1 National Institute of Nutrition and Seafood Research (NIFES) , Bergen , Norway
2 Department of Biology, High Technology Centre, University of Bergen , Bergen , Norway
3 Institute of Marine Research, Austevoll Research Station , Storebø , Norway
4 Nofima AS , Sunndalsøra , Norway
Song Linsheng
5 Present address: Norsildmel AS, Fyllingsdalen, Norway

Electronic publication date: 2013 Feb 19
Publication date: 2013
Volume: 1
Electronic Location ID: e20
Received 2012 Nov 20; Accepted 2013 Jan 8
Copyright: © 2013 Penglase et al.
Copyright year: 2013
Copyright holder: Penglase et al.
License: This is an open access article distributed under the terms of the Creative Commons Attribution License, which permits unrestricted use, distribution, and reproduction in any medium, provided the original author and source are credited.
License URL: https://creativecommons.org/licenses/by/3.0/

Keywords: Fish larvae, Cod larvae, Rotifers, Iodine requirement, Iodine toxicity, Thyroid hormones, Thyroid follicles, Mineral interactions, Colloid goitre

Funding: Norwegian Research Council This work was financed by the Norwegian Research Council (project no. 185006/S40). The funders had no role in study design, data collection and analysis, decision to publish, or preparation of the manuscript.

==============================
Copepods as feed promote better growth and development in marine fish larvae than rotifers. However, unlike rotifers, copepods contain several minerals such as iodine (I), at potentially toxic levels. Iodine is an essential trace element and both under and over supply of I can inhibit the production of the I containing thyroid hormones. It is unknown whether marine fish larvae require copepod levels of I or if mechanisms are present that prevent I toxicity. In this study, larval Atlantic cod (Gadus morhua) were fed rotifers enriched to intermediate (26 mg I kg-1 dry weight; MI group) or copepod (129 mg I kg-1 DW; HI group) I levels and compared to cod larvae fed control rotifers (0.6 mg I kg-1 DW). Larval I concentrations were increased by 3 (MI) and 7 (HI) fold compared to controls during the rotifer feeding period. No differences in growth were observed, but the HI diet increased thyroid follicle colloid to epithelium ratios, and affected the essential element concentrations of larvae compared to the other groups. The thyroid follicle morphology in the HI larvae is typical of colloid goitre, a condition resulting from excessive I intake, even though whole body I levels were below those found previously in copepod fed cod larvae. This is the first observation of dietary induced I toxicity in fish, and suggests I toxicity may be determined to a greater extent by bioavailability and nutrient interactions than by total body I concentrations in fish larvae. Rotifers with 0.6 mg I kg-1 DW appeared sufficient to prevent gross signs of I deficiency in cod larvae reared with continuous water exchange, while modelling of cod larvae versus rotifer I levels suggests that optimum I levels in rotifers for cod larvae is 3.5 mg I kg-1 DW.

Introduction

Iodine (I) is essential for vertebrates where it is utilised by the thyroid follicles to produce I containing thyroid hormones, thyroxine (T4) and tri-iodothyronine (T3) (Sutija & Joss, 2005). Iodine deficiency can lead to thyroid enlargement, termed goitre (Beckett et al., 1993; Vanderpas, 2006; Maier et al., 2007), and alter circulating thyroid hormone levels and ratios (Ruz et al., 1999). These changes are regarded as part of a compensation mechanism. Thyroid enlargement increases the capacity for thyroid hormone production, while changes in thyroid hormones levels and ratios normally favour an increase or maintenance in the circulating levels of the active form, T3, at the expense of the largely inactive T4 (Vanderpas, 2006). Paradoxically, excessive I intake can also negatively affect thyroid hormone production and produce goitre, termed I or colloid goitre (Vanderpas, 2006; Xu et al., 2006; Yang et al., 2006; Franke et al., 2008). This effect is called the Wolff–Chaikoff phenomenom (Wolff & Chaikoff, 1948) and probably occurs because concentrations of iodinated lipids in thyroid follicles increases linearly with available I (Pereira et al., 1990), and these iodinated lipids can inhibit the H2O2 production required for thyroid hormone synthesis (Ohayon et al., 1994; Panneels et al., 1994).

Thyroid hormones influence gene expression in virtually all tissues and play important roles in mediating cellular metabolism and normal development (Soldin, O’Mara & Aschner, 2008). When compensation mechanisms cannot maintain thyroid hormone homeostasis numerous metabolic and development processes can be negatively affected. For example, decreased growth, mental retardation, reduced egg hatchability, increased mortality, and decreased fertility have been observed in terrestrial vertebrates fed insufficient (Potter et al., 1982; Ferri et al., 2003; Sancha et al., 2004; Vanderpas, 2006; Robertson, Friend & King, 2008; Dong et al., 2011) or excessive I (Paulikova et al., 2002; Baker, Parr & Augspurger, 2003; Baker, 2004).

Little is known about I nutrition or toxicity in fish. Inadequate I nutrition may be prevalent in the larvae of numerous marine fish species raised in captivity. For example, feeding Senegalese sole (Solea senegalensis) larvae I enriched rotifers and Artemia prevented the development of goitre and increased growth and survival (Ribeiro et al., 2011; Ribeiro et al., 2012), cod larvae survival increased when fed I and selenium enriched rotifers (Hamre et al., 2008a), and improved growth and survival in Pacific threadfin (Polydactylus sexfilis) larvae was linked to higher levels of I in the form of iodide (I−) versus iodate (IO3−) in the rearing water (Witt et al., 2009). Additional I supplementation may improve growth and development in marine fish larvae because their requirements may mirror the higher levels of I found in their natural feed, copepods, versus the levels found in rotifers and Artemia commonly used as live feed in captivity. For example, rotifer I contents (0.6–8 mg kg-1 dry weight) are 6 to 600 fold lower than copepod levels which range from 50 to 350 mg kg- 1 DW (Hamre et al., 2008b; Moren, Sloth & Hamre, 2008). Cod larvae fed copepods, like most marine fish larvae, grow and develop better than when fed rotifers (Imsland et al., 2006; Busch et al., 2010; Koedijk et al., 2010). This has been linked to differences in the nutritional content between rotifers and copepods, particularly the differences in fatty acid profiles (Rainuzzo, Reitan & Olsen, 1997; Rodriguez et al., 1997; Park et al., 2006), but may also be related to the difference in mineral contents, of which I is the most extreme (Hamre et al., 2008b).

While several studies (Hamre et al., 2008a; Penglase et al., 2010) have indicated that I nutrition may be a critical determinate for thyroid hormone levels, ratios and survival in cod larvae, currently none has shown this conclusively. The aim of this study was to conclude if rotifers enriched up to copepod levels of I affect the thyroid status, health and growth of larval cod. Diets consisting of control rotifers containing 0.6 mg I kg- 1 DW or treatment rotifers containing either 26 mg I kg- 1 DW (MI+rotifers) or 129 mg I kg- 1 DW (HI+rotifers) were fed to cod larvae from 4 to 39 dph. The length, weight, survival, whole body mineral, thyroid hormones, thyroid follicle number and volume were measured in larval cod during this rotifer feeding period. Cod were reared on identical diets from 40 until 124 dph, and then sampled for analysis of growth and skeletal deformities. Counter to the original hypothesis, we found that cod larvae fed rotifers enriched to copepod levels of iodine displayed symptoms of iodine toxicity.

Material and methods

Cod larvae rearing

The experiment was performed at the Institute of Marine Research (IMR), Austevoll Research Station, Norway. This study was carried out within the Norwegian animal welfare act guidelines (code 750.000) at an approved facility. As this trial was assumed to be a nutrition trial based on all available studies up to the date of the trial, no specific permit was required under the guidelines. Naturally spawned and fertilised Atlantic cod eggs were obtained from in house second generation brood stock. Prior to incubation, eggs were disinfected with 200 mg L- 1 glutaraldehyde for 9 min at 6 °C and eggs were incubated with a standard protocol as described in Penglase et al. (2010). Upon 100% hatching (16 dpf, 99 day degrees), larval density in the incubators was measured via tube sampling and ranged from 2000 to 3000 larvae L−1. Three days post 100% hatching (dph), larvae (65 ± 2 µg DW fish−1, n = 2 where n is a pool of 428 or 520 fish, and 4.9 ± 0.2 mm fish- 1, n = 30 (mean ± SD), measured 5 dph) were transferred into the experimental tanks.

Larvae were stocked at an estimated density of 50 000 larvae (120 larvae L−1) in each of the nine 500 L (400 L water volume) experimental tanks (n = 3), using volumes of larvae taken from incubators based on the initial larval density measurements. The larval tanks, including colour, material, water inlets, filter size, cleaning procedures and algal additions were as described previously (Penglase et al., 2010). Water inflow to each tank (temperature 8.0 °C, salinity 34.8 ± 0.2‰, 20- µm sand/lamella filtered, degassed, from 160 m depth) started at 0.8 L min- 1 at larval transfer and increased over time to reach 4 L min- 1 by 39 dph. The water temperature in larval tanks at transfer was 8.0 °C (3 dph), and gradually increased and then maintained at 11.5 °C from 27 dph. Oxygen saturation (75–102%) and temperature were measured once daily in the outlet pipe of each tank. Dim light was provided continuously.

Rotifer culture and enrichment

Rotifers (Brachionus plicatilis. ‘Cayman’, adult lorica length 184 ± 10 µm, width 134 ± 11 µm) were batch cultured in 500 L tanks and washed as previously described (Penglase et al., 2011) (Section 2.2.5 and 2.2.6) with the exception that algae paste (Chlorella sp., Docosa, SV12, Japan) was used as the culture feed. After washing, rotifers were enriched with either a control or treatment diet. The control enrichment was 250 mg Ori-green (Skretting, Norway) million- 1 rotifers. The treatment enrichment (I+rotifers) was as per controls, but in addition 60 mg L- 1 of sodium iodide (NaI; VWR, Belgium art. no. 27915.297) was added to the water at the start of rotifer enrichment. Ori-green was prepared to manufacturer’s directions, while NaI was dissolved in cold tap water, prior to addition to rotifer enrichment tanks. Rotifers were enriched for 2 h at densities between 1000 to 2000 mL- 1 in water (as for rotifer culturing) with continual aeration and oxygenation (oxygen saturation was kept above 80%). After 1.5 h of this enrichment, an antibacterial (Pyceze, Novartis, Switzerland) was added to both control and treatment enrichment tanks at a rate of 0.2 ml L- 1. Pyceze was used to lower rotifer bacterial numbers and thus control for any antibacterial effect of I enrichment. After enrichment, rotifers were washed, concentrated to 2000–4500 rotifers mL- 1, transferred to storage tanks with aeration, and cooled rapidly (<10 min) to 8.5 °C. To maintain I concentrations in the I+rotifers, 60 mg I L- 1 (as NaI) was added to the treatment rotifer holding tank. Rotifers samples for element analysis were collected from rotifer storage tanks on 4 separate days during the larvae feeding trial. Rotifers were collected on 62 µm mesh, washed for 5 min with 12 °C saltwater, placed in 25 mL containers and stored at −20 °C.

Larval cod feeding trial

The feeding trial started at 4 dph, using rotifers as prepared in Section 2.2. Larvae were fed control or I+rotifers (HI+rotifers) or a mixture of both (80:20, control:I+, MI+rotifers). Each tank received increasing quantities of rotifers with time, starting from 3.5 million rotifers tank- 1 day- 1 at 4 dph increasing to 6 million rotifers tank- 1 day- 1 by 39 dph. The same quantity of rotifers was fed to each tank, and larvae were assumed to be fed to satiation. The rotifers were fed daily to larvae in two batch feedings of equal rotifer quantity at 10:00 and 15:00. Rotifers were poured gently into larval tanks in a circular motion to ensure even rotifer distribution and minimal larvae disturbance. Control larvae and HI+larvae were fed only control or treatment rotifers respectively. The MI+larvae were fed I+ and control rotifers at 10:00, and only control rotifers at 15:00 resulting in the overall feeding ration consisting of 80:20 control: I+rotifers.

For later analysis of skeletal deformities, fish were reared on identical diets from 40 to 124 dph. Larvae were co-fed Artemia (OK performance cysts, INVE, Belgium) and rotifers from 40 to 44 dph. Both Artemia and rotifers were enriched with Ori-green as per directions. Fish were fed only Artemia from 45 to 68 dph, co-fed Artemia and formulated diet (AgloNorse-EX1, Trofi, Tromsø, Norway) from 69 to 91 dph. Only formulated feed was fed from 92 dph (EX1; 92–94 dph, EX1 and 2; 95–103 dph, EX2; 104–115 dph, EX3; 116–124 dph). Formulated feed was administered continuously for 24 hrs day- 1 by belt feeders. Flow rate was increased from 4 L min- 1 at 30 dph to 8 L min- 1 at 108 dph, while water current speed in tanks was minimized by letting water enter through a 32 mm diameter inlet tube. Along with the increased water flow rate, oxygen saturation (64–96%) was maintained by removal of fingerling cod when required.

Sampling

Larvae were sampled for weight and length at 5, 9, 19, 30, and 124 dph, element and thyroid hormone analysis at 5, 9, 19 and 30 dph, thyroid follicle morphology at 19, 30 and 37 dph and for analysis of skeletal deformities at 124 dph. All fish were euthanised with an overdose of tricaine methane sulfonate (MS 222) upon sampling.

Larvae sampled for weight, minerals and thyroid hormones were collected on mesh (62 µm), briefly rinsed with ddH20, and patted dry from underneath with paper towel. Larvae were then placed in pre weighed tubes and frozen immediately in liquid nitrogen and stored at −80 °C until analysis. All tubes were then reweighed to determine sample wet weights. Tubes sampled for weight determination were thawed and larvae were counted (n = 20–100) to determine wet weight per larvae. Dry weight was determined from dry matter, which in turn was determined from tubes weighed before and after lyophilising. The standard length of the larvae was measured according to Hamre et al. (2008a) on 10 larvae tank- 1. Larval densities in tanks were measured at 30 dph as described by Penglase et al. (2010).

Larvae for investigation of thyroid follicle morphology (3 fish tank- 1) were selectively sampled to be similar in length, thus representing similar levels of morphological development (Sæle & Pittman, 2010). Larvae were placed in individual tubes containing 1 ml of 4% paraformaldehyde in PBS buffer at pH 7.2. Samples were left overnight and then transferred to separate tubes containing 70% ethanol, where they remained at 4 °C until embedding.

For analysis of skeletal deformities, cod juveniles (124 dph, n = 50 per tank) were measured for length and weight, frozen flat and subsequently stored in individual labelled plastic bags at −20 °C until analysis. Survival in one HI+ tank decreased to zero prior to this sampling, so data represents the mean ± SD n = 2 for the HI+fish at 124 dph.

Analytical methods

Mineral analysis

Samples for analysis of total I were digested under alkaline conditions using tetra methyl ammonium hydroxide ((CH3)4NOH, Tamapure-AA, Tama chemicals, Japan) and then analysed with ICP-MS (Agilent 7500 series, USA) as described by Julshamn, Dahl & Eckhoff (2001) with cod muscle (BCR-422, Belgium) used as the standard reference material. Samples for the analysis of other elements were prepared by wet digestion with nitric acid (65% HNO3 Suprapur®, Merck, Germany) and hydrogen peroxide (30% H2O2, Merck, Germany), in a microwave (Ethos 1600, Milestone, USA) as described by Julshamn et al. (2004). The samples were then analysed with ICP-MS along with blanks and standard reference material as described previously (Julshamn et al., 2004), with modifications to the mass of Mn (Mass 55) and Pb (Mass 208) measured. The standard reference materials used (NIST-SRM 1566, Oyster tissue, USA; TORT-2, NRC, lobster hepatopancreas, Canada) had similar concentrations of minerals as the samples analysed.

Thyroid hormone analysis

Thyroid hormones extraction from larvae was carried out according to Einarsdottir et al. (2006) with some modifications. Approximately 500 mg WW of larvae (440 to 630 mg) was homogenised (Precellys 24, Bertin technologies, France) in 1 ml of ice-cold methanol (Sigma-Aldrich, Germany), and then stored over night at −20 °C. Samples were then centrifuged (30 min, 3000 rpm, 4 °C) and the supernatant removed from the pellet. This extraction procedure was then repeated on the pellet twice more. Nitrogen was used to dry the supernatant of methanol. Lipids were removed from samples by modified Folch extraction. To eliminate any lipid in samples, the dried extracts were dissolved in barbital buffer (0.1 M pH 8.6), methanol and chloroform (1:1:2). The aqueous phase containing T4 and T3 was transferred to a fresh tube, evaporated with nitrogen and frozen at −20 °C until use. To estimate extraction efficiency, ≈1000 cpm of [ 125I]-rT3 (NEX109, Perkin Elmer, USA) were added to the homogenates after homogenisation. The extraction efficiency ranged between 71 and 81%. The T4 and T3 content were determined by radioimmunoassay using an external standard curve according to Einarsdottir et al. (2006), and further corrected for the extraction efficiency of each sample. Larval T3 contents were also reanalysed with a DELFIA® T3 Kit (PerkinElmer, Turku, Finland) according to manufacturer’s instructions.

Thyroid follicle histology

Larvae were dehydrated in an increasing gradient of ethanol and embedded in Technovit 7100 as per directions (Heraeus Kulzer, Wehrheim, Germany). The resin blocks were then sectioned into 5 µm thick slices, and every second section was placed on a slide and stained with toluidine blue. The follicle number within the pharyngeal region was counted and the area of epithelium and colloid were measured for each larvae (2 larvae tank- 1 at 19 and 30 dph, 1 larvae tank- 1 at 37 dph) at 200× magnification using a microscope and computer assisted program CAST-grid version 2 (Olympus, Albertslund, Denmark) according to Sæle et al. (2003). The colloid and epithelium volume were calculated using area and width of sections, and skipped sections were assumed to have the same volume as that measured on the preceding section. Larval sections were also scanned for evidence of thyroid follicles in the kidneys, as has been observed for some other fish species such as common carp (Cyprinus carpio) (Geven et al., 2007), but none were observed.

Radiography

The radiographic imaging and analysis of skeletal deformities was performed as described previously (Penglase et al., 2010) with skeletal deformities and degree of deformities classified according to Baeverfjord et al.. Briefly, radiographic images were visually examined for any skeletal pathology, and deviations from normal were recorded. Deviations were classified in axial deviations, vertebral deviations and head deformities, and further classified into sub categories and degrees of severity.

Calculations

Larval survival was adjusted using a linear individual probability timeline for each tank to calculate the probable survival of sampled larvae had they remained in the tanks until density measurements at 30 dph, using the following equation; Estimated survival of sampled larvae at time point Y=((100−((100−S)/T1)∗T2)/100)∗C2

where S is the measured survival % at 30 dph, T1 equals the time period in days from the start (5 dph) and end (30 dph) survival measurements (25 d), T2 equals the number of days time point Y is from the end survival measurement and X equals the number of larvae sampled at time point Y. This equation was used to produce two numbers, one for the sampling at 9 (T2 = 21) and one at 19 (T2 = 11) dph, and along with the number of larvae sampled on day 30, were added to the larvae measured in tanks from density measurements taken after sampling at 30 dph. Specific growth rates (SGR) of cod larvae were calculated with the following equation SGR =  (e∧((lnW1−lnW0)/(t2−t1))−1) × 100 where W0 and W1 are the initial and final dry weights (tank means) respectively, and t2–t1 is the time interval in days between age t1 and t2 (Ricker, 1958). Fulton’s condition factor (FC) was calculated using FC = Weight (g)∗100/Length (cm)3. The I concentration ratio (CR) between larvae and feed was CR = Larval I content (mg kg- 1 DW)/rotifer I content (mg  kg- 1 DW).

Data analyses

Statistica software (Statsoft Inc., 2008, Tulsa, USA, Ver.9) was used for statistical analysis of data except when GraphPad Prism (GraphPad Software, San Diego, CA, USA, Ver. 5) was used to model fit the I concentration of cod larvae versus rotifers, and on data from 124 dph, as the loss of one replicate in the HI larvae group prevented ANOVA analysis between all three groups. Data analysed with Statistica were checked for homogeneity of variances using Levene’s test before having significance tested with one-way ANOVA followed with Fisher’s least significant difference (LSD) homogeneity post-hoc test at each time point. Data with significantly different variance between treatments according to Levene’s test (p < 0.05) was log transformed before analysis. As the density of cod larvae has a large effect on growth (Koedijk et al., 2010), growth data during the larval stage was analysed with ANCOVA with the final larval density in tanks included as a continuous predictor. Data from 124 dph were analysed using regression, and tested against the null hypothesis that rotifer I content had no effect on outcome. Differences among means were considered significant at p < 0.05.

Results

Cod larvae growth

No statistically significant differences in cod larvae length (p > 0.08) or dry weight (p > 0.25) occurred between treatments, although high variation between tanks may have masked effects (Fig. 1). On average, cod larvae grew from 4.9 ± 0.2 mm to 6.8 ± 0.5 mm in length, and 0.065 ± 0.002 mg to 0.27 ± 0.07 mg fish- 1 in dry weight from 5 to 30 dph (Fig. 1), representing a specific growth rate of 6.3% day- 1 during this period.

Figure 1 Cod larvae length and dry weight.

Length (Data set L; mm fish-1, left y axis) and dry weight (Data set W; mg fish-1, right y axis) of cod larvae fed control (□), MI (○) or HI (●) rotifers, from 5 to 30 dph. At 5 dph, data are mean ± SD of 2 analytical parallels for dry weight and mean ± SD (n = 30) for length. Data at all other dph are mean ± SD (n = 3) where n represents the average of 10 larvae tank-1 measured for length, and a group of 47 to 520 larvae group weighed then counted to determine individual mass.

Survival

There were no statistical differences (p > 0.39) in the cod larval survival adjusted for sampled larvae (see Section 2.6) between treatments at 30 dph which were 28 ± 2, 39 ± 20 and 30 ± 15% for controls, MI and HI groups respectively, while the average for all groups was 32 ± 13%. Survivals based solely on densities in tanks at 30 dph without taking into account sampled larvae were 12 ± 2, 19 ± 12 and 12 ± 10% for controls, MI and HI groups respectively.

Iodine and other essential element concentrations in rotifers and cod larvae

Control rotifers contained 0.60 ± 0.33 mg I kg- 1 DW while HI+rotifers contained 129 ± 101 mg I kg- 1 DW (Table 1). Whole body I levels in cod larvae were significantly different between groups (p < 0.01, Fig. 2). Cod larvae (5 dph) had a starting concentration of 4.0 ± 0.3 mg I kg- 1 DW. Between 9 and 30 dph average I concentrations were 1.6 ± 0.3 mg I kg-1 DW for control larvae, while MI larvae had 3 fold higher levels (4.9 ±2.4 mg I kg- 1 DW), and HI larvae 7 fold higher levels (11.0 ± 3.3 mg I kg-1 DW) than controls. Other element concentrations were also affected by treatment in both rotifers and cod larvae. HI+rotifers contained less Fe and Mn than controls (Table 1), while HI larvae contained more Mn, Fe and Cu than controls and MI larvae at one or more time points (Figs. 2b–2c and 2e). Both HI and MI larvae contained higher levels of Co than controls (Fig. 2d), while larval Zn and Se concentrations were unaffected by treatment (Figs. 2f–2g). Rotifer macro mineral concentrations were unaffected by treatment (Table 1), but increased levels of Ca and Mg, and lower levels of P and K were observed in HI larvae in comparison to controls and/or MI larvae during the rotifer feeding period (Fig. 3).

Figure 2 Essential micro element concentration in whole cod larvae.

Essential micro element concentrations (mg kg-1 DW) in whole cod larvae fed either control (□), MI (○) or HI (●) rotifers, from 5 to 30 dph. Letters denote statistically significant differences in mineral concentrations between treatments at a given day (one-way ANOVA; p < 0.05). Data are mean ± SD (n = 3), except at 5 dph where data are mean ± SD of analytical parallels.

Figure 3 Essential macro element concentrations in whole cod larvae

Essential macro mineral concentrations (g kg−1 DW) in whole cod larvae fed either control (□), MI (○) or HI (●) rotifers, from 5 to 30 dph. Letters denote statistically significant differences in mineral concentrations between treatments at a given day (one-way ANOVA, p < 0.05). Data are mean ± SD (n = 3), except at 5 dph where data are from a single analysis.

Table 1 Essential element concentrations (mean ± SD) in control and sodium iodide (HI+) enriched rotifers.

Mineral	Control rotifers	HI+rotifers	P value	
Microminerals (mg kg−1 DW, n = 4)	
Iodine	0.60 ± 0.33a	129 ± 101b	0.04*	
Manganese	9.0 ± 0.6a	8.0 ± 0.4b	0.03*	
Copper	10.8 ± 6.6	12.1 ± 6.2	0.78	
Zinc	41 ± 2	38 ± 4	0.24	
Iron	150 ± 10a	135 ± 3b	0.04*	
Cobalt	0.17 ± 0.07	0.16 ± 0.06	0.85	
Selenium	0.03 ± 0.01	0.04 ± 0.02	0.76	
Macrominerals (g kg−1 DW, n = 3)	
Calcium	2.2 ± 0.2	2.6 ± 0.3	0.16	
Phosphorus	12.3 ± 0.6	11.1 ± 0.8	0.13	
Magnesium	5.5 ± 0.8	7.0 ± 0.9	0.09	
Potassium	12.0 ± 0.2	11.7 ± 0.3	0.31	
Notes.

* Superscript letters denote statistically significant differences between rotifer groups (one-way ANOVA p < 0.05).

Iodine concentration in cod larvae versus rotifers

The rate of increase in cod larvae I concentrations decreased as dietary I levels (rotifer I concentration) increased, and thus the I concentration ratio between cod larvae and rotifers displayed a negative trend (Fig. 4; p < 0.01). The age of the cod larvae did not effect their I concentration ratio (p = 0.96). The model predicts that the ratio of I in cod larvae versus rotifers equals 1 when rotifers have 3.5 mg I kg- 1 DW (Fig. 4).

Figure 4 Cod larvae iodine concentration in relation to their feed.

Ratio of iodine concentration (mg kg-1 DW) in cod larvae versus their diet (rotifer iodine levels (mg kg-1 DW)). X axis is log transformed. Line represents best fit model (Morrison Ki, R2 = 0.94). Data are mean ± SD (n = 9).

Thyroid hormones and thyroid follicle morphology

There were no differences in thyroid hormone levels or ratios between treatments (Fig. 5). Data was normalised to aid interpretation, as T3 results were higher than obtained previously (Penglase et al., 2010) and the high result was validated by the analysis of T3 with a second method (see methods). The total volume of thyroid follicle was 1.3 fold lower and the total epithelium volume was 1.4 fold lower per fish in HI versus MI larvae, but not controls, at 30 dph (Figs. 6a and 6c). The thyroid follicle colloid to epithelium ratio was higher in HI larvae than controls at 19 (1.7 fold) and 37 (1.8 fold) dph, while MI larvae did not differ from controls (Fig. 6d). No statistically significant differences were observed between groups in colloid volume or total number of thyroid follicles (Figs. 6b and 6e). Images of thyroid follicle sections from control and HI larvae at 37 dph are shown in Fig. 7.

Figure 5 Cod larvae thyroid hormone levels and ratios.

Normalised mean thyroid hormone levels (NML) in cod larvae fed either control (□), MI (○) or HI (●) rotifers. Graph A is tri-iodothyronine (T3), Graph B is thyroxine (T4), while graph C is the ratio between the NML of T3/T4. Data are mean ± SD (n = 3) for all data points except controls at 30 dph which has an outlier removed in graph B and C (n = 2).

Figure 6 Cod larvae thyroid follicle morphology.

Thyroid follicle morphology in cod larvae fed either control (□), MI (○) or HI (●) rotifers. Graph A shows the total number of follicles per fish, Graph B is the total thyroid follicle volume per fish, Graphs C and D show the volume of colloid or epithelium per fish, Graph E shows the ratio between the colloid and epithelium volumes. Letters denote statistically significant differences between treatments per time point (one-way ANOVA, p < 0.05). Data are mean ± SD (n = 3) where n consists of the average measurements from two fish per tank at 19 and 30 dph, and one fish per tank at 37 dph.

Figure 7 Thyroid follicle sections from cod larvae.

Thyroid follicle section from cod larvae (37 dph) fed either control (A) or HI (B) rotifers. Sections are stained with toulidine blue. C; thyroid follicle colloid, E; example of thyroid follicle epithelium. Scale bars are 100 µm.

Weight, length, condition factor and rate of skeletal deformities at 124 dph

There were no significant differences in the weights (average; 2.50 ± 0.19 g), lengths (6.14 ± 0.16 cm) or condition factors (1.05 ± 0.03) between groups at 124 dph (Table 2, n = 400). Neck axis angle became closer to normal (180 ± 3 degrees) with increasing I levels in rotifers, but there were no differences in any of the other skeletal deformity measurements (Table 2).

Table 2 The weight, length, condition factor and percent of skeletal deformities in 124 dph cod fed control, MI or HI rotifers from 5 to 39 dph and identical diets thereafter.

		Control	MI	HI	P value	
Growth	Weight (g)	2.49 ± 0.22	2.56 ± 0.18	2.44 ± 0.25	0.67	
	Length (cm)	6.12 ± 0.17	6.19 ± 0.19	6.09 ± 0.18	0.75	
	Condition factor	1.06 ± 0.05	1.05 ± 0.01	1.04 ± 0.01	0.43	
Skeletal	Lower jaw	4.0 ± 2.0	9.3 ± 7.6	5.3 ± 4.7	0.96	
deformitiesa *	Short upper jaw	3.3 ± 3.1	0	0	0.26	
	Palate bone	1.3 ± 1.2	0	3.0 ± 4.2	0.28	
	Neck axis average (degrees)b	186 ± 0.4	186 ± 0.6	184 ± 1.0	0.01	
	Neck axis <183 degrees	25 ± 2	23 ± 10	39 ± 12	0.08	
	Fused vertebrate	26 ± 6	15 ± 5	29 ± 10	0.45	
	No. of affected vertebrae (average fish- 1)	2.1 ± 0.2	2.6 ± 0.6	2.6 ± 0.6	0.39	
	Scoliosis	17 ± 2	21 ± 7	15 ± 4	0.42	
	Back axis	1.3 ± 1.2	1.3 ± 1.2	3.1 ± 1.3	0.10	
	Total fish with malformation	58 ± 5	52 ± 7	64 ± 11	0.26	
Notes.

Data are mean ± SD (n = 3) except for HI data which are n = 2.

a % of population unless otherwise indicated.

b Normal neck angle is 180 ± 3°(Baeverfjord et al.).

* Data analysed using regression (p < 0.05).

Discussion

The hypothesis of this study was that commercially reared cod larvae fed rotifers would benefit from increased dietary I. The bases for this hypothesis were, one; rotifer I concentrations are often at the lower end or below juvenile/adult fish requirements (NRC, 2011), two; rotifers have 6–600 fold lower concentrations of I than copepods (Hamre et al., 2008b), the natural feed of cod larvae (Thompson & Harrop, 1991), three; cod larvae have better growth and development when fed copepods versus rotifers (Busch et al., 2010; Koedijk et al., 2010), four; I levels in copepod fed cod larvae are higher than rotifer fed cod larvae (Busch et al., 2010), and five; increased growth and/or survival has been observed in larval stages of several marine fish species fed or reared in environments with increased levels of bioavailable I (Hamre et al., 2008a; Witt et al., 2009; Ribeiro et al., 2011; Ribeiro et al., 2012).

However, in contrast to the hypothesis, the increased thyroid follicle colloid to epithelium (C/E) ratio observed in this study indicates that I toxicity occurred in cod larvae fed rotifers with 129 mg I kg- 1 DW. This observation purely in relation to the I level is not unexpected. A high C/E ratio in thyroid follicles is a classic symptom of I induced toxicity termed I (Baker, 2004) or colloid goitre, and occurs in mice at dietary I concentrations 10 fold higher than requirements with increasing severity developing with increasing I ingestion rates (Yang et al., 2006). What is interesting is that despite copepods containing 50–350 mg I kg- 1 DW (Solbakken et al., 2002; Hamre et al., 2008b), cod larvae have either similar or lower thyroid follicle C/E ratios when fed copepods compared to rotifers (Grøtan, 2005). Furthermore, I concentrations observed in cod larvae fed natural zooplankton (29 mg I  kg-1 DW at 27 dph; (Busch et al., 2010)) were 2.2 fold higher than the highest level observed in the current study (HI larvae, 30 dph; 13 ± 4 mg I kg- 1 DW). Thus it appears that high I concentrations in copepods do not induce morphological changes in thyroid follicles consistent with I toxicity, but do appear to be effectively transferred from copepods to fish larvae upon consumption.

It is possible that copepods do not induce I toxicity in fish larvae due to nutrient interactions. For example, I toxicity can be prevented by the simultaneous presence of the bromine anion (B r−) in animals ranging from chicks (Gallus gallus) (Baker, Parr & Augspurger, 2003) to Artemia (S Penglase et al., unpublished data). The exact mechanism for this B r−/I− interaction is still unknown, but it has been demonstrated that B r− decreases iodide accumulation in the thyroid follicles and increases I excretion via the kidneys in rats (Pavelka, 2004). Although the bromide concentrations in whole copepods and rotifers are unknown, we speculate that copepods have relatively high levels of bromide reflecting the high levels found in other marine organisms, and this bromide helps prevent I toxicity in fish larvae. In the marine ecosystem, bromine is naturally found at similar high concentrations as I in seaweed (Romaris-Hortas, Moreda-Pineiro & Bermejo-Barrera, 2009), adult fish (Arafa et al., 2000; Wan et al., 2010), and as part of the hard chitin structures of crustaceans such as crabs (Cribb et al., 2009; Schofield et al., 2009) and copepods (Perry, Grime & Watt, 1988).

Thyroid hormone levels and ratios were similar between cod larvae groups, and this is probably due to the compensatory changes observed in the thyroid follicles. For example pathological changes of over 70% in thyroid gland morphology have been observed in dogs (Canis lupus familiaris) with little change in circulating TH levels (Graham, Refsal & Nachreiner, 2007), and significant changes in fish thyroid follicle morphology with little change in thyroid hormone levels have also been reported for fish (Hawkyard et al., 2011; Morris et al., 2011).

Few differences were found between the control and MI larvae groups, with the exception of whole body I concentrations. The increased whole body I content of cod larvae demonstrates the effective transfer of I from the rotifers to the cod larvae. The current study differs to previous studies exploring the uptake of I in fish larvae, as it attempted to ensure the ingestion of graded levels of I by maintaining the I concentration in the prey organism up until the point of feeding. Srivastava et al. (2012) found that I leaches rapidly from rotifers after enrichment with sodium iodide. Previous studies have found no difference in the I concentration of cod larvae fed control or I supplemented rotifers (Hamre et al., 2008a; S Penglase et al., unpublished data), and this is probably a consequence of I leaching from rotifers in the minimum 1.5 to 2 h period between rotifer enrichment and feeding of the rotifers to cod larvae in these studies.

In the current study, cod larvae iodine level increases were proportionally smaller for each increase in dietary I levels; control fish were 2.7 fold higher, while MI were 5 fold lower and HI larvae were 12 fold lower in I than their respective diets. Body stores of minerals are a good indicator of status (Baker, 1986), and the decreasing level of I retention in cod larvae relative to feed I levels indicates that requirements were met at levels lower than those fed to MI larvae. Modelling of the ratio between cod larvae and rotifer I concentrations predicts that based on a ratio of 1:1, rotifer I concentrations of 3.5 mg kg- 1 DW meet cod larvae requirements. Both food and water contribute to the I status of adult, juvenile (Lall, 2002) and larval fish (Witt et al., 2009; Ribeiro et al., 2011). Alongside the I content in the continuously exchanging seawater (88 µg I L- 1, Moren, Sloth & Hamre, 2008), the control rotifers in the current study with 0.6 mg I kg- 1 DW appeared to prevent any gross signs of I deficiency in cod larvae. This is at the lower end of the 0.6–1.1 mg I kg- 1 DW recommended by the national research council (NRC, 2011) as the I requirements of juvenile/adult fish.

The reason that symptoms of severe I deficiency such as classic goitre have been observed in other fish studies is probably due to water parameters. Clear signs of I deficiency (goitre, decreased growth and/or decreased survival) occurred in fish larvae reared in either recirculation systems (Ribeiro et al., 2011; Ribeiro et al., 2012) or well water (Witt et al., 2009). Nitrate (NO3−) is goitrogenic as it competitively blocks iodide uptake by the sodium iodide symporter (Tonacchera et al., 2004), and NO3− at levels commonly found in recirculation systems causes goitre in sharks (Crow et al., 1998; Morris et al., 2011). Furthermore, ozone (O3) used as a disinfectant in recirculation systems readily oxidises bioavailable I, iodide (I−) to iodate (IO3−) (Sherrill, Whitaker & Wong, 2004). Dissolved iodate is presumed to have low bioavailablity for fish (Sherrill, Whitaker & Wong, 2004), and higher levels of iodate compared to iodide were correlated to poor growth and survival in well water reared pacific threadfin larvae (Witt et al., 2009). Thus in recirculation systems, the presence of high levels of goitrogens (NO3−) and low levels of dissolved bioavailable I (I−) may increase the dietary I requirements of fish larvae over those reared with continuous water exchange where nitrate and its precursors are continuously removed and iodide continuously replaced, such as in the current study.

Along with thyroid follicle morphology, dietary I also influenced the mineral composition of cod larvae. While most of the tested mineral concentrations in MI larvae were similar to controls, HI larvae had 10 to 25% higher levels of Ca, Mg, Mn, Fe, Co and Cu and around 10% lower levels of P and K than controls at one or more time points within the rotifer feeding period. For most of the minerals, differences in levels were observed by the first sampling point (9 dph; 4 days of feeding on rotifers). The differences cannot be explained by the feed; the HI rotifers had ≈10% less Mn and Fe, and no statistical differences were observed in Ca, Mg, K, P, Cu or Co concentrations. Hamre et al. (2008a) found that cod larvae fed increased levels of both I and Se had an 8% increase in whole body copper levels, similar to this study. Nguyen et al. (2008) found increased or decreased copper levels (20%) in red sea bream (Pagrus major) larvae depending on whether they were fed rotifers enriched with Mn alone or alongside Zn. While it is known that I deficiency can alter mineral distribution and homeostasis of Cu, Mn, Fe and Zn (Giray et al., 2003), to our knowledge this is the first data demonstrating that I oversupply can also effect mineral homeostasis. Although there were few differences in growth or skeletal deformities observed between treatments at 124 dph, there was a small but statistically significant improvement in the neck axis angle in the HI compared to the control and MI cod groups (Table 2), and this may be linked to the differences in cod mineral concentrations in the larval stage.

Conclusion

Iodine enriched rotifers increased the levels of I in cod larvae, although as I levels in rotifers increased the increases in cod larvae I levels became proportionally smaller. Few differences occurred between cod larvae reared on control diets with 0.6 mg I kg- 1 DW and those reared on diets with 26 mg I kg- 1 DW, while the I concentration ratio between cod larvae and rotifers suggests cod larvae have an I requirement of 3.5 mg I kg- 1 rotifers DW. Rotifers with copepod levels of I (129 mg I kg- 1 DW) changed cod larvae whole body concentration of many essential minerals and induced changes in thyroid follicles morphology consistent with colloid goitre. The data presents one of the first observations of dietary induced I toxicity in fish, and suggests that I toxicity in fish larvae may be determined to a greater extent by I bioavailability and nutrient interactions than by body burdens of I.

This work was financed by the Norwegian Research Council (project no. 185006/S40). Thank you to technical staff at IMR Austevoll and NIFES for fish husbandry, sampling help and skilled analytical assistance, especially Stig Ove Utskot and Berit Solli. Thank you to Karin Pittman for allowing access to equipment for thyroid follicle morphology quantification.

Additional Information and Declarations

Competing Interests

Author Contributions

Animal Ethics

Kristin Hamre is an Academic Editor for PeerJ. Otherwise, there are no conflicts of interest connected to this manuscript.

S Penglase and K Hamre conceived and designed the experiments, performed the experiments, analyzed the data, contributed reagents/materials/analysis tools and wrote the paper.

T Harboe conceived and designed the experiments, performed the experiments, contributed reagents/materials/analysis tools and wrote the paper.

Ø Sæle and S Helland analyzed the data, contributed reagents/materials/analysis tools and wrote the paper.

A Nordgreen conceived and designed the experiments, analyzed the data and wrote the paper.

The following information was supplied relating to ethical approvals (i.e. approving body and any reference numbers):

The study was performed undere the Norwegian animal welfare act guidelines (code 750.000) at the Institute of Marine Research, which is an approved facility. As this trial was assumed to be a nutrition trial based on all available studies up to the date of the trial, no specific permit was required under the guidelines.

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
