# Peer review of "Iodine nutrition and toxicity in Atlantic cod (Gadus morhua) larvae"

_PeerJ, doi:10.7717/peerj.20_

## Round 0.1 · original submission · Minor Revisions

The manuscript is well written and orgnized. The experiments were conducted in an appropriate manner and data were moderately well presented. It is potentially interesting for researches on iodine nutrition and toxicity in cod larvae. As the reviewers suggested, some tables and figures in the manuscript should be improved.

·

Basic reporting

This is a well-written and timely article based on a robust experimental design and thorough analyses. Unfortunately the manuscript mentions two Tables that were not included in the files I received. The figures are all relevant although the figure legends could have been more specific about how many separate figures were meant to be embraced by the figure legend (please insert something like "Fig. 2 A-G" at the beginning of the legend). Otherwise the manuscript is of very high intellectual and technical standards.

Minor point: line 226-228 uses the word sample 4 times.

Experimental design

The experimental design was clearly stated and reproducible by those wishing to expend the energy. It was clearly suitable for the question. All standards and codes of conduct have been adhered to.

The description of the radiography and especially measurement of the neck angle is too brief to be useful (1.5 lines), although the results of this analysis are mentioned in Discussion.

I also had a small question about how the number of follicles was counted (lookup sections?).

Only one other question arose in my mind, regarding the admixture of rotifers from Control and High iodine enrichments to create a Medium Iodine diet. I can see no suggestion of selectivity by the cod larvae, which would have resulted in high variations for that group, so this may not be important at all.

Validity of the findings

The findings are comprehensively given and analysed. The mortality of one tank changed some of the statistical analyses but this is a minor point. The findings are justified and are put into several relevant contexts: that of dietary requirements and interactions, that of source of water used to rear the larvae (recirculation, wellwater or seawater) and that of longterm effects.

It might be interesting to have a line or two about synthesis of thyroid hormones in other organs than the thyroid follicles (it happens in fish) and what the "gross signs" of iodine deficiency would be in larvae (line 377).

Additional comments

Nice work. I like the clear statement about iodine requirements and the codicil about nutrient interactions.

It would be useful to have the two missing tables, therefore I am recommending "Minor revisions".

Reviewer 2 ·

Basic reporting

the manuscript is well written and organized properly

Experimental design

yes

Validity of the findings

yes

Additional comments

The manuscript describes effect of Iodine on Atlantic cod (Gadus morhua) larvae. Overall, the manuscript is well written and organized properly, I think the manuscript is suitable for the journal of “The PeerJ” at current status. Herein, the comments and suggestions are given below in an attempt to improve the paper.
1) In the abstract section, the sentence “larval Atlantic cod (Gadus morhua) were fed rotifers enriched to intermediate (26 mg I kg -1 dry weight; MI group) or copepod (129 mg I kg -1 DW; HI group) I levels and compared to cod larvae fed control rotifers (0.6mg I kg -1 DW).” should be revised. I could not find any evidence you used copepod for feed trial in Material and method section.
2) the manuscript contained so many Figures, I recommend the author should be integrated some of them together.

---

## Round 0.2 · accepted · Accept

The manuscript is well revised and can be accepted for PeerJ. Thanks.